# Long-Chain Polyunsaturated Fatty Acids n−3 (n−3 LC-PUFA) as Phospholipids or Triglycerides Influence on *Epinephelus marginatus* Juvenile Fatty Acid Profile and Liver Morphophysiology

**DOI:** 10.3390/ani12080951

**Published:** 2022-04-07

**Authors:** Paulo H. de Mello, Bruno C. Araujo, Victor H. Marques, Giovana S. Branco, Renato M. Honji, Renata G. Moreira, Artur N. Rombenso, Maria C. Portella

**Affiliations:** 1Beacon Development Company, King Abdullah University of Science and Technology (KAUST), Thuwal 23955-6900, Saudi Arabia; 2Centro de Aquicultura da Unesp (CAUNESP), Universidade Estadual Paulista (Unesp), Jaboticabal 14884-900, Brazil; maria.c.portella@unesp.br; 3Departamento de Fisiologia, Instituto de Biociências, Universidade de São Paulo, São Paulo 05508-090, Brazil; victor.marqueslorenti@gmail.com (V.H.M.); giovana_branco@hotmail.com (G.S.B.); renata.fish@gmail.com (R.G.M.); 4Centro de Biologia Marinha, Universidade de São Paulo, São Sebastião 11600-000, Brazil; bruno.araujo@cawthron.org.nz (B.C.A.); honjijp@gmail.com (R.M.H.); 5Cawthron Institute, Nelson 7042, New Zealand; 6CSIRO, Livestock and Aquaculture Program, Bribie Island Research Centre, Woorim, QLD 4507, Australia; artur.rombenso@csiro.au

**Keywords:** dusky grouper, eicosapentaenoic acid, docosahexaenoic acid, lipid metabolism, marine phospholipid

## Abstract

**Simple Summary:**

Fish feeding is responsible for almost 60% of the total cost of marine fish production, so nutritional studies are always in demand. Dusky grouper is a promising species for aquaculture, but studies of the species’ nutrition are still lacking. The present study investigated the effects of fatty acids, provided as different lipid classes in four different diets, for *Epinephelus marginatus* fingerlings and evaluated their physiological and morphological responses in order to provide new insights about grouper nutrition. The fatty acids, provided as different lipid classes, modified the physiological and morphological responses of *E. marginatus*, showing that the inclusion of these lipids as different classes should be considered in order to obtain better results in terms of fish fillet quality. These results provided valuable insights into the nutrition and physiology of dusky groupers, helping to pave the way for the establishment of this fish species as a produced species worldwide.

**Abstract:**

Phospholipids (PL) are membrane components composed of fatty acids (FA), while triglycerides (TG) are a main source of energy and essential FA. Polyunsaturated FA (PUFA), such as docosahexaenoic acid (DHA), and eicosapentaenoic acid (EPA), are essential for marine carnivorous fish; thus, an 8-week experiment was performed to evaluate the influence of DHA and EPA, provided as PL and TG, on the morphophysiology of *Epinephelus marginatus* juveniles. A basal diet was manufactured, and DHA and EPA in PL form (PL1—low amount PL2—high amount) and TG form (TG1—low amount; TG2—high amount) were added. Dusky grouper juveniles were equally distributed in 12 tanks of 20 animals each, and liver and muscle were sampled for metabolic analysis. The total hepatic lipids in PL1 and PL2 were higher when compared to the initial, TG1 and TG2 groups. Total lipids in muscle were higher in PL2 and TG1 than PL1 and TG2, respectively. Diets rich in DHA and EPA in PL and TG resulted in higher deposition of these FA in the muscle polar fraction. However, fish fed diets containing lower amounts of DHA and EPA in PL and TG stored those in the muscle neutral fraction and liver, centralizing the storage of DHA and EPA.

## 1. Introduction

*Epinephelus marginatus*, popularly known as dusky grouper, is a marine fish species with economic and ecological importance in many places where it occurs [1]. It has good potential for production and preservation of natural stocks since it is considered vulnerable, according to the International Union for Conservation of Nature and Natural Resources (IUCN) red list [2,3]. However, a main concern is the lack of understanding of the nutrition requirements of this species [4].

Lipids represent one of the most structurally and functionally diverse groups of biomolecules [5] and, among these, the importance of phospholipids (PL) is recognized and well-established for many animal groups, such as fish [6]. PLs are precursors of a variety of biologically active mediators of metabolism, such as diacylglycerols, inositol phosphate, and platelet-activating factors [6]. They are also important components of biological membranes, providing important molecules as substrates for membranes, such as choline, inositol, phosphorus, essential fatty acids (EFA) and energy. They also act as lipid emulsifiers (surfactant) [6]. Several studies have reported the positive effects of dietary PL supplementation, both in freshwater and marine species, such as ayu (*Plecoglossus altivelis*) [7], striped beakfish (*Oplegnathus fasciatus*) [8], shrimp (*Marsupenaeus japonicas*) [9], fingerlings of rainbow trout (*Oncorhynchus mykiss*) [10], fingerlings of Atlantic salmon (*Salmo salar*) [10], barred sorubim juveniles (*Pseudoplatystoma fasciatum*) [11], and brown trout fingerlings (*Salmo trutta*) [12].

Another important class of lipids are the triglycerides (TG), which act as the main energy source in fish, being catabolized to provide metabolic energy for maintenance and development, in addition to providing EFAs, acting in membrane synthesis, and acting as important metabolism modulators and chemical signals [13]. The FA can be divided into long-chain polyunsaturated fatty acids (LC-PUFA), such as docosahexaenoic acid (DHA 22:6n−3), eicosapentaenoic acid (EPA 20:5n−3), and arachidonic acid (ARA 20:4n−6), which are considered EFA [13].

The dietary inclusion of DHA and EPA for marine fish larvae in captivity is essential and requirement studies have been conducted for over 30 years [14,15,16]. There are several positive effects resulting from the inclusion of these EFA in the diet for marine fish species, such as increased growth and survival. In addition, EFA are involved in several immune system responses, since they act in the synthesis of eicosanoids such as prostaglandins and leukotrienes [13,17].

The aforementioned FA are present in both TG and PL, and feed manufacturing procedures influence several metabolic processes throughout development. *Dicentrarchus labrax* larvae fed with DHA and EPA in the PL form showed significantly higher growth performance, survival, development, and maturation of digestive organs in relation to larvae fed with the same FA in TG, suggesting that these larvae had a greater ability to efficiently use these FA in PL form [18].

The liver plays a significant role in metabolism, working on glycogen and lipid metabolism and storage. The liver’s morphology reveals the nutritional status, and it is an essential tissue used to evaluate physiological condition, reflecting whether the diet is appropriate and/or whether there are any feed inadequacies or restrictions [19]. The liver’s morphological aspects accurately reflect nutritional status, and these changes can be evaluated by hepatocyte size, cell volume, hepatic glycogen, and lipid content [19,20].

Due the importance of these FA classes to marine fish species, the present study aimed to investigate the influence of dietary n−3 LC-PUFA (DHA and EPA) supplementation in PL and TG form on the lipid metabolism, morphology of the liver, and expression of genes related to the lipid metabolism of dusky grouper juveniles for 8 weeks in order to evaluate its effects on growth, survival, and fish tissue incorporation.

## 2. Materials and Methods

### 2.1. Diet Manufacturing

Four diets with different DHA and EPA levels in the form of TG (2 diets, group TG 1 with lower TG levels and group TG2 with higher TG levels) and PL (2 diets, PL 1 with lower PL levels and PL2 with higher PL levels) were designed (Table 1).

Additionally, Table 2 shows the fatty acid profile of the diets.

Diets were manufactured at the Oceanographic Institute from São Paulo University (USP) in Ubatuba-SP according to in-house protocol. Each of the 4 diets was tested in the different groups in triplicate (total of 12 tanks, 3 tanks per treatment). Morphometric data and sampling were performed at the beginning and after 60 days of feeding.

### 2.2. Experimental Design

All procedures in the present study were conducted and authorized according to the animal ethics committee of the USP and the National Council for the Control of Animal Experimentation (CONCEA) (n° 337/2018), where the experiment was carried out. Two hundred and fifty-two dusky grouper juveniles of approximately 1 g were acquired from Redemar Alevinos (Ilha Bela, São Paulo, Brazil), and transferred to the Marine Biology Center of University of São Paulo (CEBIMar), where the experiment was carried out. Fish were maintained for one week in 2000 L tanks (temperature 25.0 ± 0.5 °C; oxygen 6.52 ± 1.2 mg/L^−1^ and total ammonia < 0.05 mg/L^−1^) and hand fed twice a day with 5% of total weight/day with a commercially available marine fish diet.

Then, the animals were transferred to experimental tanks with a volume of 500 L each in a recirculation aquaculture system (RAS) equipped with UV, biological filter, mechanical filter, pump and water temperature control. Fish were randomly divided into 4 experimental groups in triplicate, containing 20 animals with 1.43 ± 0.22 g per tank, totaling 60 animals per group and 240 animals in total. Fish were hand fed twice a day (9:00 am and 5:00 pm) until apparent satiation. An initial sampling of 12 animals was carried out before the start of the experiment (initial group (IG)) and a final sampling was carried out at the end of the experiment, at which point 4 animals per triplicate were sampled, totalizing 12 animals sampled per treatment. Temperature and dissolved oxygen were monitored daily (YSI model 55, YSI Inc., Yellow Springs, OH, USA) and total ammonia and nitrate were monitored every 7 days (API test kits, Mars Fishcare Inc., Chalfont, PA, USA) (temperature 26 ± 0.5 °C; oxygen 6.52 ± 1.2 mg/L^−1^ and total ammonia and nitrate values < 0.05 mg/L^−1^).

For tissue collection, the animals were euthanized in benzocaine (1 g diluted in 10 L) and an aliquot of the liver and muscle were obtained to analyze total lipid content, fatty acid profile, and the expression of genes related to fatty acid metabolism. A portion of the liver was fixed in 4% buffered formalin for histological analysis. For the metabolic and molecular analyses, the tissues were immediately frozen in liquid nitrogen and transferred to an ultra-freezer (−80 °C).

### 2.3. Total Lipids and Fatty Acid Profile

For evaluation of the liver and muscle tissue total lipids and fatty acids profiles, the method of Folch et al. [21] was used. For quantification, the method of Frings et al. [22] was used. To evaluate the polar and neutral fractions of the diets, liver and muscle, the lipid extracts of samples were separated using an activated silica column protocol from Yang (1995). The acetyl chloride methylation procedure (5% of HCl in methanol) [23] was used to produce the methyl ester in the neutral and polar fractions. To determine the FA profile, we used a gas chromatograph, Scion 436, coupled with a flame ionizer (FID) and CP8410 autosampler. A CP Wax capillary column (30 m length, 0.25 μm thickness, 0.25 mm inner diameter) was also used, with hydrogen as the carrier gas, and a 1.4 mL/min cm/s of linear velocity. The program of the column was: 1 min at 170 °C, a 2.5 °C/min ramp to 240 °C, then hold for 5 min. The FID temperature was 260 °C and the injector temperature was 250 °C. The FA acid percentage was based in the area of the peak when compared to a standard (Supelco, 37 components; Sigma-Aldrich).

## 3. Gene Expression

### 3.1. RNA Extraction and cDNA Synthesis

The total RNA extraction of the tissues was performed using TRIzol Reagent (Sigma-Aldrich, St. Louis, MO, USA). The quality and concentration was accessed using a NanoDrop One Spectrophotometer (Thermofisher, Waltham, MA, USA) and the samples used showed an A_260/280_ ratio higher than 1.7. After that, we diluted the samples and added DNase (AM1907 Invitrogen™ Thermofisher, Waltham, MA, USA).

A reverse transcription enzyme was used for the RNA (1 µg) to cDNA reaction (18064014 Invitrogen™ Thermofisher, Waltham, MA, USA). To validate the effectiveness of the reaction, we performed samples with RNA without adding the enzyme to check the transcription (negative controls).

### 3.2. Primer’s Validation

Primers for lipid synthesis and β-oxidation genes in groupers were previously designed and validated by Araujo et al., [24] (Table 3). The housekeeping gene *elongation factor 1 alpha* (*ef1a*) was used, and had already been validated for this species [25]. All primers were synthesized by Thermofisher (Invitrogen™, Waltham, MA, USA).

To confirm the specificity of the primers, a conventional PCR was performed with a cDNA sample pool with each primer pair and subjected to electrophoresis with 1% agarose gel to visualize the specific bands by excitation with UV light.

### 3.3. Real-Time Quantitative PCR (qPCR)

To measure the primers’ efficiency, we considered only those with R^2^ near 1 and a range between 90–100% of efficiency. To evaluate the gene expression in the grouper liver samples (*n* = 6), we used the 2^−(ΔΔCt)^ method with Step One Real-Time PCR System (Applied Biosystems). After that, we determined the cycle thresholds and used the *ef1a* gene and IG group to normalize the data.

### 3.4. Histology Analysis

Fish liver tissues were fixed in 4% buffered formalin solution and then transferred to 70% ethanol. For processing, samples were dehydrated in increasing ethanol concentrations from 70–96% and embedded in methacrylate (Technovit 7100, Heraeus Kulzer, Germany). A microtome (Leica HistoCore AutoCut, Wetzlar, Germany) was used to obtain 2 μm serial sections and a glass knife of 10 mm was used. Sections were stained with hematoxylin and eosin (Sigma-Aldrich). Liver samples were examined under a Leica DM1000 LED light microscope and the images were captured with a computerized image capture system (Leica MC170HD photography camera; Leica LAS Interactive Measurements, Wetzlar, Germany).

### 3.5. Statistical Analysis

The data were expressed as the mean ± SD (standard deviation of the mean). Normality was verified using the Shapiro–Wilks test. The comparison between groups (variable: diet) was performed with a one-way ANOVA followed by Tukey’s post hoc test. GraphPad Prism 5.01 (GraphPad Software Inc., San Diego, CA, USA) was used for gene expression data and SigmaStat software for Windows version 3.5 (SigmaStat Software, San Diego, CA, USA) was used for total protein, total lipids and fatty acids profiles. For all analyses, *p* < 0.05 was considered significant.

## 4. Results

### 4.1. Survival and Morphophysiological Parameters

The performance results and biological indexes are presented in Table 4.

Body weight, final weight, and weight gain did not change across the experimental groups, and no mortality was observed in all experimental groups. The liver of animals from PL2 and TG2 groups showed significantly higher lipid deposition compared to the IG (PL2—*p* = 0.002; TG2—*p* = 0.002) and TG1 groups (PL2—*p* = 0.003; TG2—*p* = 0.002), with the PL1 and TG1 (*p* = 0.075) groups having similar levels compared to IG (PL1—*p* = 0.065; TG1—*p* = 0.937). Experimental diets increased lipid concentration in the muscles of animals from PL2 and TG1 groups, which were significantly higher than those from the IG (PL2—*p* = 0.001; TG1—*p* = 0.008), PL1 (PL2—*p* = 0.001; TG1—*p* = 0.008), and TG2 groups (PL2—*p* = 0.001; TG1—*p* = 0.001). The TG2 group exhibited lower levels than the IG (*p* = 0.015), but similar levels to the PL1 group (*p* = 0.596).

### 4.2. Fatty Acid Profile

The percentage of saturated FA (SFA) was lower in the experimental groups, compared to the animals from the IG (PL2—*p* = 0.003; TG1—*p* = 0.001; TG—*p* = 0.001), with the exception of the PL1 group (*p* = 0.086), which showed similar levels (Table 5).

The SFA content in the animals from PL2 and PL1 (*p* = 0.486) groups was similar to and higher than the TG1 (PL 2—*p* = 0.001; PL1—*p* = 0.001) and TG2 groups (PL2—*p* = 0.029; PL1—*p* = 0.013), respectively; the latter was lower than TG1. The percentage of MUFA was higher in the hepatic neutral fraction of animals from all experimental groups compared to IG (PL1—*p* = 0.029; PL2—*p* = 0.001; TG1—*p* = 0.029; TG2—*p* = 0.001), with the PL2 group showing the highest percentages, followed by the TG2 and PL1 (*p* = 0.614) groups, which were similar to each other. The TG1 group displayed the lowest MUFA percentage. The PUFA content decreased in the animals from the experimental groups in relation to the IG (PL1—*p* = 0.029; PL2—*p* = 0.029; TG2—*p* = 0.001), except for the TG1 group (*p* = 0.343).

The percentage of n−3 PUFAs decreased in the neutral hepatic fractions of animals from PL1, PL2, and TG2 groups in relation to the IG (PL1—*p* = 0.029; PL2—*p* = 0.029; TG2—*p* = 0.029). On the other hand, it increased in the animals from the TG1 group (*p* = 0.016). The n−6 PUFAs were significantly less deposited in the neutral hepatic fractions of all experimental groups, compared to IG (PL1—*p* = 0.001; PL2—*p* = 0.001; TG1—*p* = 0.016; TG2—*p* = 0.029). Animals from the PL2 group presented lower n−6 PUFA percentages in their neutral hepatic fractions, while animals from PL1 (*p* = 0.029) exhibited higher percentages than the PL2 (*p* = 0.029) and TG1 groups (0.050). The n3/n6 PUFA ratio showed a significant decrease in the experimental groups in relation to the IG (PL1—*p* = 0.029; PL2—*p* = 0.029), except for TG1 (*p* = 0.016), which increased the percentage in this ratio, and TG2, which was similar (*p* = 0.343).

Most groups showed similar SFA content in the polar hepatic fractions to the IG (Table 6), with the exception of the PL2 group (*p* = 0.029), which presented lower percentages.

The SFA percentages of the polar hepatic fractions of the TG1 and TG2 groups were higher than the PL1 (TG1—*p* = 0.042; TG2—*p* = 0.026) and PL2 groups (TG1—*p* = 0.029; TG2—*p* = 0.029), which also differed from each other (PL1 × PL2—*p* = 0.025). The MUFA percentages were higher in the hepatic polar fractions in the animals from all experimental groups, compared to the IG (PL1—*p* = 0.016; PL2—*p* = 0.001; TG1—*p* = 0.016; TG2—*p* = 0.029). Fish from the TG2 group had the highest MUFA percentage, followed by PL2, PL1 and TG1, the last two being similar in their deposition percentages. The PUFA percentages were lower in the hepatic polar fractions of fish from all experimental groups, in relation to the IG (PL1—*p* = 0.021; PL2—*p* = 0.001; TG1—*p* = 0.016; TG2—*p* = 0.001), with the TG2 group presenting the lowest percentage of this FA class. The FA with the highest percentages were 22:6n−3 and 18:2n−6.

Among the n−3 PUFAs, most of the groups demonstrated a significant decrease in relation to the IG (PL1—*p* = 0.001; PL2—*p* = 0.001; TG1—*p* = 0.001), with the exception of TG1 (*p* = 0.410), which showed similar values to the IG. The animals from the PL1 group presented higher percentages of n−3 PUFA compared to the PL2 (PL1—*p* = 0.004) and TG2 groups (PL1—*p* = 0.016), but similar values to the TG1 group (PL1—*p* = 0.261). The percentages of n−6 PUFA were higher in the polar fractions of fish from the PL2 group and lower in the TG1 and TG2 groups in relation to the IG (PL2—*p* = 0.002; TG1—*p* = 0.001; TG2—*p* = 0.001). Within the experimental groups, the percentages of n−6 PUFA were highest in PL2, followed by PL1 (PL2 = 0.003), TG1 (PL2—*p* = 0.001; PL1—*p* = 0.001), and TG2 (PL1—*p* = 0.001; PL2—*p* = 0.001; TG1—*p* = 0.030); they were different among all groups. The n−3/n−6 PUFA ratio displayed a significant decrease in most groups compared to the IG (PL1—*p* = 0.004; PL2—*p* = 0.001; TG2—*p* = 0.001); however, the TG1 group (*p* = 0.016) showed a significant increase in this ratio.

The SFA percentages increased in the neutral fractions of the muscles of the animals from the PL1 and PL2 groups compared with the IG (PL1—*p* = 0.001; PL2—*p* = 0.001), but did not change in the TG1 (*p* = 0.123) and TG2 groups (*p* = 0.247). In fact, they were lower than the PL1 (TG1—*p* = 0.001; TG2—*p* = 0.001) and PL2 groups (TG1—*p* = 0.001; TG2—*p* = 0.001) (Table 7).

MUFA increased in the experimental groups in relation to the IG (PL1—*p* = 0.001; PL2—*p* = 0.001; TG1—*p* = 0.001; TG2—*p* = 0.001). The percentages of MUFA in the neutral fractions of the muscles in fish from the PL2 and TG2 groups (*p* = 0.339) were similar, and higher than PL1 (PL2—*p* = 0.001; TG2—*p* = 0.001) and TG1 (PL2—*p* = 0.001; TG2—*p* = 0.001); this percentage was higher in animals from TG1 compared to PL1 (*p* = 0.026). PUFA percentages were lower in the neutral fractions of the muscles in fish from all experimental groups compared to IG (PL1—*p* = 0.001; PL2—*p* = 0.001; TG1—*p* = 0.001; TG2—*p* = 0.001). Additionally, all groups were different from each other; the TG1 group showed the highest percentages, followed by the TG2 (PL1—*p* = 0.001; PL2—*p* = 0.001; TG1—*p* = 0.025), PL1 (PL2—*p* = 0.002; TG1—*p* = 0.001; TG2—*p* = 0.001), and PL2 groups (PL1—*p* = 0.002; TG1—*p* = 0.001; TG2—*p* = 0.001). The main PUFA were 18:2n−6 and 22:6n−3. The n−3 PUFA percentages were lower in the neutral fractions of muscles in fish from most experimental groups in relation to the IG (PL1—*p* = 0.001; PL2—*p* = 0.001; TG2—*p* = 0.004), with the exception of the TG1 group (*p* = 0.247), which presented similar percentages. Similarly, the TG1 group displayed the highest percentages, followed by the TG2 (*p* = 0.001), PL1 (*p* = 0.001), and PL2 groups (*p* = 0.001). The percentages of n−6 PUFA in the neutral fractions were lower in fish from all experimental groups in relation to the IG (PL1—*p* = 0.001; PL2—*p* = 0.024; TG1—*p* = 0.001; TG2—*p* = 0.001), with the PL1 group presenting higher percentages than all experimental groups (PL2—*p* = 0.006; TG1—*p* = 0.001; TG2—*p* = 0.001). The n−3/n−6 PUFA ratio was lower in the experimental groups than in IG (PL1—*p* = 0.001; PL2—*p* = 0.001; TG1—*p* = 0.001), except for TG2 (*p* = 0.203). Additionally, all groups differed from each other, with the TG1 (PL1—*p* = 0.001; PL2—*p* = 0.001; TG2—*p* = 0.001) group showing the highest ratio, followed by the TG2 (PL1—*p* = 0.001; PL2—*p* = 0.001; TG1—*p* = 0.001), PL1 (PL2—*p* = 0.001; TG1—*p* = 0.001; TG2—*p* = 0.001), and PL2 (TG1—*p* = 0.001; TG2—*p* = 0.001) groups.

The SFA percentages increased in the polar fractions of the muscles of fish from PL1 and TG1 compared to IG (PL1—*p* = 0.010; TG1—*p* = 0.010), but did not change in the PL2 (*p* = 0.200) and TG2 groups (*p* = 0.666) (Table 8).

The MUFA percentage increased in most experimental groups compared with IG (PL2—*p* = 0.043; TG1—*p* = 0.010; TG2—*p* = 0.01), except PL1 (*p* = 0.257). Comparing the experimental groups, the percentages of MUFA were lower in the polar fractions of the muscles from PL1 fish compared to PL2 (*p* = 0.001), TG1 (*p* = 0.001), and TG2 (0.001) fish. The PUFA percentages decreased in fish from PL1, TG1, and TG2, compared the IG (PL1—*p* = 0.001; TG1—*p* = 0.001; TG2—*p* = 0.028). The PUFA percentages of the PL2 group were higher than the PL1 (*p* = 0.001), TG1 (*p* = 0.001), and TG2 groups (*p* = 0.030), with 22:6n−3 and 18:2n−6 more representative than other FA. The percentages of n−3 PUFAs decreased in most groups in relation to the IG (PL1—*p* = 0.010; TG1—*p* = 0.010; TG2—*p* = 0.039), with the exception of the PL2 group (*p* = 0.308). Fish from the TG1 group presented the lowest n−3 PUFA percentage compared to the other experimental groups, while PL2 presented the highest percentage. The percentages of n−6 PUFAs were lower in PL1 and TG1 compared to the IG (PL1—*p* = 0.001; TG1—*p* = 0.001), and higher in PL2 (*p* = 0.033). Between the experimental groups, the percentage was the highest in PL2 and lower in PL1 and TG1. The n−3/n−6 ratio decreased in the TG1 and TG2 groups in comparison to IG (TG1—*p* = 0.047; TG2—*p* = 0.049).

### 4.3. Gene Expression

Data of the relative expression of elongase 5 (*elovl5*) (a), carnitine palmitoyltransferase (*cpt-1*) (b), acetyl CoA carboxylase (*acc*) (c), acyl CoA dehydrogenase (*acadvl*) (d), acyl-CoA Oxidase (*acox*) (e), fatty acid synthase (*fas*) (f) and stearoyl CoA desaturase (delta 9) (*scd*) (g) are presented in Figure 1.

In relation to the genes of FA synthesis, fish fed the PL1 diet presented hyperregulation of the gene *elovl5* in comparison to those fed the TG2 diet. However, fish fed the TG2 diet presented upregulation of the *fas* in comparison to fish fed the PL2 diet. Finally, fish fed the PL1 diet presented hyperregulation of the *scd* in comparison to those fed PL2 and TG2. In relation to FA oxidation genes, fish fed the PL2 diet presented hyperregulation of the *cpt-1* in comparison to fish fed the TG2 diet, while fish fed the TG1 and PL1 diets had no differences. The same pattern was observed for the *acadvl*. There were no differences in the relative expression of the *acc* and *acox*.

### 4.4. Liver Morphology

Liver morphology was assessed to evaluate possible different patterns of the feeding in the liver. PL1 (Figure 2a) hepatocytes were smaller in size than the rest of the groups. PL2 (Figure 2b) exhibited the largest hepatocytes among all groups with large cytoplasm containing high lipid contents and low protein contents. TG1 (Figure 2c) and PL1 exhibited low vacuolization levels, as opposed to TG2 (Figure 2d) which, as PL2, exhibited high vacuolization levels with larger hepatocytes.

## 5. Discussion

First, approaching morphometric parameters, the diets enriched with DHA and EPA in the form of PL and TG showed similar results in juvenile dusky groupers, suggesting, at this life stage, a need for adequate amounts of DHA and EPA rather than specific requirements for those in the PL or TG form. The effects of offering EFA as PL on diets have been studied mainly in the larval stages [6], while the results with juveniles differ according to the species. Rainbow trout (*O. mykiss)* larvae increased growth, while no difference was observed in juveniles (average weight of 0.5 g) fed with PL [26]. Atlantic salmon (*S. salar)* fingerlings showed a lack of positive growth response driven by PL supplementation after reaching a certain body mass (7.5 g) [10]. As opposed to the results observed in our experiment with dusky grouper juveniles and the aforementioned studies, juveniles of hybrid striped bass (*Morone chrysops* × *M. saxatilis*) [27], Pacific tuna (*Thunnus orientalis*) juveniles [28], and cobia (*Rachycentron canadum*) juveniles [29] exhibited higher growth when fed with DHA and EPA-rich diets in the PL form. Similarly, higher growth was observed in the larvae of other marine fish species fed with DHA and EPA in the PL form, such as European sea bass (*D. labrax*) [18,30], Pacific tuna (*Thunnus orientalis*) [28], sea bream (*Sparus aurata*) [31], and meagre (*Argyrosomus regius*) [32].

Dusky groupers demand high levels of DHA and EPA during the initial phase of development [33]; thus, specific diets are essential and have been investigated with the aim of developing a complete and adequate diet for this promising commercial species [24]. It is worth remembering that each species has its own EFA requirements, but the diets formulated for the present study met the known nutritional requirements recommended for marine species, such as European sea bass (*Dicentrarchus labrax*), turbot (*Scopthalmus maximus*), and halibut (*Hipoglossus hipoglossus*), providing percentages of DHA/EPA around 2:1 [16]. The DHA and EPA content in PL and TG resulted in better survival in European bass larvae [18], as well as in hybrid striped bass larvae [27]. On contrary, juvenile European bass [30] and meagre larvae (*A. regius*) [34] showed no differences in survival, suggesting that responses to diets rich in DHA and EPA in PL and TG are species-specific and dependent on the developmental phase [6].

The total hepatic lipids showed a different deposition pattern between the experimental groups, with the PL2 and PL1 groups exhibiting concentrations between 1.2 and 2 times higher than the IG, TG1, and TG2 groups. Therefore, the diets provided energy and structural sources stored as lipids in the liver tissue. The TG1 group presented the same concentrations of total hepatic lipids as the IG, suggesting that other tissues (such as muscle) may be important storage tissues. This was deemed plausible, as the amount of lipids in the muscle was higher in animals from the TG1 and PL2 groups. Lipid metabolism is species-specific and is affected by several factors, such as life stage, temperature, diet composition, genetics, and other factors, which may reflect different deposition in tissues, showing a certain plasticity in the response of the lipids [35,36].

Regarding the liver neutral FA profile, there was no observed direct impact of the diets. In other words, the diets did not reflect their FA percentage in the liver tissue. The maintenance of high levels of SFA in the neutral fractions of the livers showed that dusky groupers could use this substrate for storage due the sufficient amount in the feed, since this FA class is generally used as a metabolic substrate [37,38]. This was observed in previous studies with this species, in which fish preferentially used SFA for β-oxidation [24], and similar results were also observed in a congeneric species (*Epinephelus malabaricus*) [39]. In the hepatic neutral fraction, the PL2 group stored higher MUFA percentages, even though the supplied diet had similar percentages between the experimental diets in both fractions. In PUFA, the opposite was observed; the PL2 group had lower percentages of this FA class and the TG1 group had higher percentages, followed by TG2. Thus, TG1 and TG2 stored the DHA and EPA, since these FAs play roles in essential biological functions [4,40]. The same was observed with n−3 PUFAs and in n−3/n−6 ratios, in which the TG1 and TG2 groups showed higher percentages of storage of these FAs in the fractions of liver, showing a potential sparing effect for these FAs. Similar results were also observed in a study that evaluated bone development and tissue FA profiles in cod (*Gadus morhua*) larvae fed DHA and EPA in TG and PL form [41].

In the liver polar fractions, an opposite pattern of neutral fraction was observed. SFA exhibited higher deposition, followed by PUFA, while MUFA showed lower percentages, with just the TG2 group having higher MUFA percentages in relation to PUFA. This profile showed a direct effect of the diet and tissue incorporation, suggesting that MUFA were preferentially oxidized [37,38], while PUFA were retained due to their physiological importance [6,40] and the biochemical characteristics of the PL molecule [42]. In addition, the percentages of DHA and EPA deposition in the liver polar fractions reached levels similar to those observed in other previous studies and the PL2, PL1 and TG1 groups, ranging from 14–20% of DHA and 3–6% of EPA, with the exception of the TG2 group, which stored around 6% of DHA and 2% of EPA. These are considered low levels, since hepatocyte PLs normally store high PUFAs amounts, as observed in other studies with dusky groupers in which percentages were between 23–32% for DHA and 6–7% for EPA [24]. A previous study investigating EPA and DHA-rich diets in PL and TG composition observed high percentages of 16:0 in the diets, stimulating PL biosynthesis in fish larvae [43]. The same occurs for humans, as observed in a study where 16:0 appeared to facilitate the incorporation of DHA into tissues [44]. This effect also occurred in meagre larvae; a positive relationship was observed between 16:0 and DHA in the PL [34]. In the present study, DHA levels were as high as 16:0 levels, which may also have had a direct relationship with the levels of FAs stored in the polar fractions of liver tissue. The n−3/n−6 ratio observed in the liver polar fractions of dusky groupers was relatively low (1 and 2.84%) when compared to a previous study (2.7 and 3.7%) with this same species [24].

In the muscle neutral fractions, it was possible to observe that the PL2 and PL1 groups stored higher SFA and MUFA percentages compared to TG1 and TG2 groups, which deposited higher MUFA percentages in the muscle. This pattern suggested that animals from PL groups catabolized SFA preferentially, while the animals from the TG group catabolized MUFA preferentially, a common profile observed in fish [45]. Since MUFA availability in the neutral fraction was higher due to the biochemical characteristics of TG, high catabolism of these FAs in white muscle has been observed in several marine fish species [37,38,45]. The DHA and EPA percentages in the experimental groups were much lower compared to the IG, but mainly in the PL2 and PL1 groups, showing that the diets had an important effect on this profile. The experimental animals catabolized the PUFAs or stored these FAs in polar fractions or other tissues. This was unusual, since the SFA and MUFA percentages in the diets were much higher, and the tendency is to preserve PUFAs instead of SAT and MUFA [37,38,45]. It is noteworthy that the TG1 and TG2 groups showed higher PUFA deposition in the muscle compared to the PL2 and PL1 groups, possibly preserving these FAs for use in advanced phases due to their physiological importance throughout development [4,6,40]. In dusky grouper (*E. marginatus*) larvae [33], PUFA levels between 3 to 6 times higher than those observed in the present study were observed in the neutral fraction, confirming that the developmental stage influences FA composition in different tissues [45]. Another relevant point observed was the difference in the n−3/n−6 ratio between the PL and TG groups, showing that retention in the neutral fractions was more effective in the group fed DHA- and EPA-rich diets in TG form—which reached 4 times this ratio in group TG1 compared to PL2.

In the muscle polar fractions, an SFA increase was observed in fish from PL1 and TG1 in relation to the other experimental groups, suggesting a retention of these FA for posterior utilization. The PUFA exhibited the largest variation in this fraction, while the PL1, TG1, and TG2 groups showed drastic reductions in comparison to the IG. Aside from this significant PUFA decrease in these experimental groups, PL2 maintained the same percentages when compared to the IG, meaning that the PL2 group reflected the DHA and EPA levels provided in the diet in the muscle, showing a direct nutritional effect. The same was observed for n−3, with higher levels observed in the PL2 group. The FA muscle composition was influenced mainly by lipid and FA composition of diets. However, other factors exist, such as species’ ability to synthetize LC-PUFA from their precursors (C18 carbons FA) [36,46]. The PL2 group reflected the FA profile provided in the diet compared to the muscle. This has been highly supported by studies [4,36,46]. Additionally, DHA is highly physiologically relevant, while EPA is expendable, especially in marine larvae [14,15]. Therefore, the PL2 diet provided high DHA levels; this group had around 14% DHA in the muscle.

The results observed with respect to *cpt-1* showed that the PL2 diet upregulated the expression of this gene. This gene, *cpt-1*, is a key enzyme that realizes the first steps in FA metabolism [5]. Several studies with marine fish demonstrated a correlation with the relative expression pattern of this gene and the FA profile of diet and tissues [47,48,49]. Similar to *cpt-1*, *acadvl* is related to FA oxidation pathways; however, it is more related to LC-PUFA catabolism. The relative expressions of *acadvl* and *cpt-1* in the present study were higher in fish fed the PL2 diet. In general, the results suggested that LC-PUFA catabolism was higher in fish fed DHA and EPA in the PL form.

It is known that when LC-PUFA are supplied in excess, they can be oxidized preferentially by fish [40,50]. Perhaps because the fish receives this substrate already in a storage form (TG) it is less energetically costly to keep it that way, while PL is easily absorbed by the fish [6] and, when it reaches physiological levels, is excessively oxidized, decreasing its availability in tissues. Nevertheless, physiological requirements for LC-PUFA are different from the requirement to maintain high LC-PUFA levels in tissues [51]. Thus, providing DHA and EPA as PL for dusky groupers at this life stage may not be interesting in terms of the maintenance of these FA in the fillets or for productive performance, since the results did not show any improvement in both cases.

As fish fed DHA and EPA as TG preserved these LC-PUFA in the tissues, these animals preferentially oxidized other substrates like SFA. The expression patterns of *fas* demonstrated that fish fed the TG2 diet upregulated the expression of this gene. This enzyme is responsible for synthesizing an important SFA, 16:0. As such, this result may be related to an energy balance regulation to compensate 16:0 loss. The *elovl5* relative expression was consonant with the tissue FA profile, as this enzyme converts C18 into C20 FA [5,45] and fish fed TG had higher percentages of long-chain C20 FA. Therefore, they may not need to synthesize these FA, since they were acquired directly from the diet. However, a trend toward downregulation of *acox* was observed in fish fed the TG2 diet, as compared to fish fed the other experimental diets.

Both groups PL2 and TG2 exhibited higher lipid deposition in the liver compared to the PL1 and TG1 groups, showing a possible vacuolization effect in the hepatic tissue that, if continuously consumed, could cause future problems—in severe cases, steatosis. High lipid accumulation can be related to high levels of 18:2n−6 (linoleic acid) in the diet, since this FA has a lipogenic effect in mammals once it increases the fatty acid synthase and glucose-6-phosphate dehydrogenase activities [52]. Additionally, the liver can be considered the main lipolytic tissue in dusky groupers, even compared to muscle tissue, playing several important roles regarding lipid deposition as TG [53]. Thus, the diet composition can have a direct effect on liver lipid stores, as is commonly observed in other marine fish species such as sea bream and sea bass [54].

## 6. Conclusions

Diets containing different DHA and EPA levels in PL and TG form did not result in significant differences in performance and biological indexes in dusky groupers at this developmental stage. Dusky groupers fed high DHA and EPA levels as PL and TG showed higher deposition of these FA in the muscle polar fraction. However, when fed with lower DHA and EPA levels as PL and TG, these essential fatty acids were preferentially stored in the muscle neutral fraction. Liver was an important tissue in DHA and EPA deposition, in addition to n−3 and n−6 PUFAs. It was possible to observe that the group fed DHA- and EPA-rich diets as TG had higher percentages of these FA in muscle, which can be considered an indicator of fillet quality from a nutritional point of view.

These results provide valuable insights on nutrition and physiological aspects of *E. marginatus*, supporting the development of specific sustainable formulations. However, additional information will be required to establish a diet that provides ideal PUFA amounts to increase fish fillet quality when reared in captivity.

## Figures and Tables

**Figure 1 animals-12-00951-f001:**
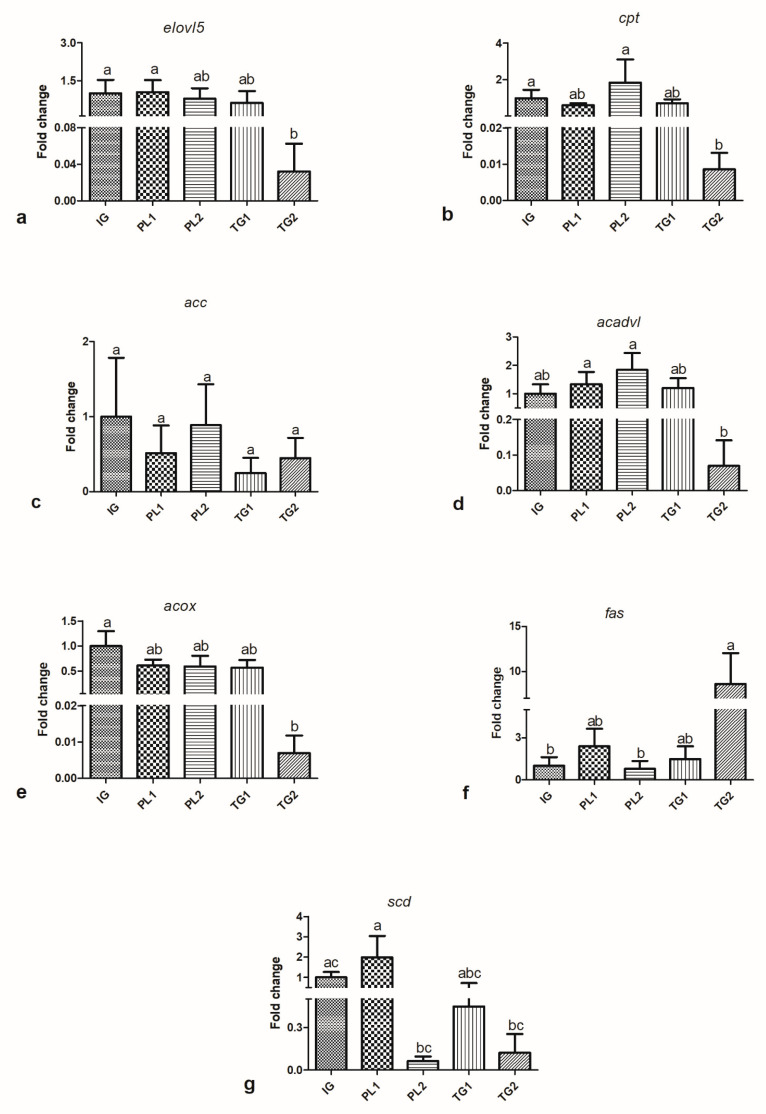
Relative expression levels of FA synthesis and β-oxidation genes (**a**) *evovl*; (**b**) *cpt*, (**c**) *acc*, (**d**) *acadvl* (**e**) *acox* (**f**) *fas* (**g**) *scd*. The values represent the mean ± standard errors (*n* = 6). The transcript level from each gene is calculated relative to the other genes’ levels using the raw cycle threshold value for each gene and then normalized against *ef1a*. The values shown are the fold induction changes relative to the average Ct value for all genes. PL1-low levels of phospholipids; PL2-high level of phospholipids; TG1-low levels of triglycerides; TG2-high levels of triglycerides. The different letters present significant (*p* < 0.05) differences among diets.

**Figure 2 animals-12-00951-f002:**
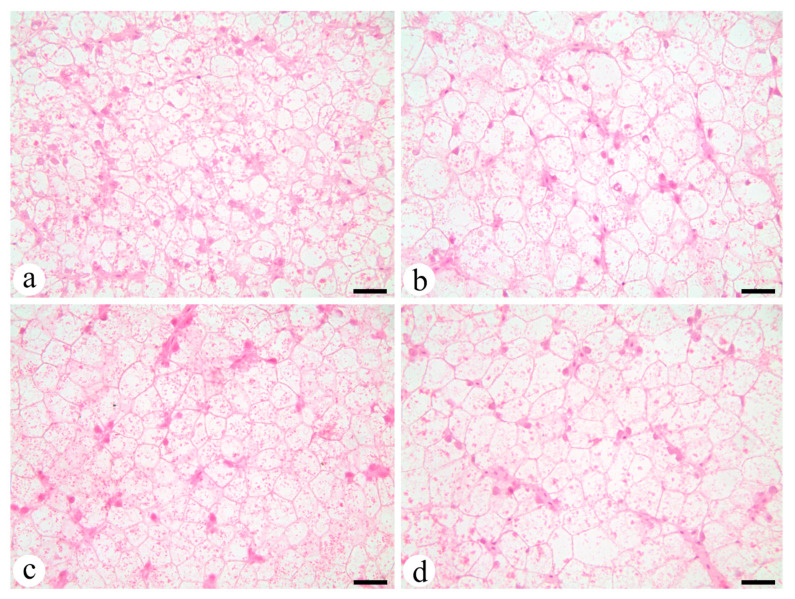
Morphologies of the hepatocytes of dusky grouper juveniles after the 8-week feeding trial. (**a**) PL1 group; (**b**) PL2 group; (**c**) TG1 group; (**d**) TG2 group. Bars 25 µm.

**Table 1 animals-12-00951-t001:** Dietary formulation and proximal composition (g kg^−1^).

	Experimental Diets (g kg^−1^)			
Ingredient	PL1	PL2	TG1	TG2
Defatted Fish Meal ^a^	450	450	450	450
Defatted Poultry Meal ^b^	68.4	68.4	68.4	68.4
Hemoglobin ^c^	30	30	30	30
Wheat Meal ^d^	300	300	300	300
Taurin ^e^	8	8	8	8
Premix ^f^	40	40	40	40
StayC ^g^	2.5	2.5	2.5	2.5
Sodium Benzoate ^h^	1	1	1	1
BHT ^i^	0.1	0.1	0.1	0.1
DHA/EPA enriched oil (Phosphotech) ^j^	0	0	10	30
Coconut Oil ^k^	45	25	50	40
Olive Oil ^l^	35	15	40	30
Marine Lecithin (Phosphotech) ^m^	20	60	0	0
Proximal Composition (g/kg)				
Total Lipids	76	108.3	94.3	117.3
Total Proteins	427.2	409.2	403.5	405.2
Ash	165.9	161.9	159.7	162.7
Moisture	77.3	81.33	78.9	78.7

^a^ and ^b^ (defatted 3× with hexan); ^f^ Premix (IU kg^−1^ or g/kg of premix): vitamin A. 2.5MIU; vitamin D3. 0.25 MIU; vitamin E. 16.7 g; vitamin K3. 1.7 g; vitamin B1. 2.5 g; vitamin B2. 4.2 g; vitamin B3. 25 g; vitamin B5. 8.3; vitamin B6. 2.0 g; vitamin B9. 0.8; vitamin B12. 0.005 g; biotin. 0.17 g; vitamin C. 75 g; colin. 166.7 g; inositol. 58.3 g; etoxiquin. 20.8 g; copper. 2.5 g; iron. 10.0 g; magnesium. 16.6 g; manganese. 15.0 g; zinc. 25.0 g; ^a,b,c,d,e,f,g,h,i^ Nutricon Ltda-Me. Sao Paulo. Brazil; ^j^ DHA e EPA enriched Oil (DHA 45% EPA 15%). Phosphotech Laboratories. ZAC de la Lorie. France.; ^k^ Copra Indústria Alimentícia Ltda. Alagoas. Brazil; ^l^ Victor Guedes. Ind. Com. S.A. Abrantes. Portugal; ^m^ Marine Lecithin. Phosphotech Laboratories. ZAC de la Lorie. France. PL1-low levels of phospholipids; PL2-high level of phospholipids; TG1-low levels of triglycerides; TG2-high levels of triglycerides.

**Table 2 animals-12-00951-t002:** Fatty acid profile of the neutral and polar fractions of the experimental diets (% of fatty acids).

FA	Experimental Diets (% as Phospholipids (PL))				Experimental Diets (% as Triglycerides (TG))			
	PL1	PL2	TG1	TG2	PL1	PL2	TG1	TG2
12:0	3.44	6.98	7.06	4.52	19.89	21.53	21.60	20.20
14:0	2.70	3.72	4.31	3.33	8.53	8.98	8.97	13.12
16:0	33.72	23.59	23.78	23.01	17.91	13.48	12.68	8.70
18:0	9.99	9.31	9.94	10.69	5.46	4.11	3.96	4.08
ΣSFA	49.86	43.59	45.08	41.56	51.79	48.11	47.20	46.11
16:1n−7	2.39	2.07	3.38	3.10	2.38	1.63	1.56	2.08
18:1n−9	18.95	18.86	22.71	20.58	33.12	38.04	38.53	33.62
18:1n−7	2.74	2.29	2.86	2.92	2.05	1.59	1.65	1.94
20:1n−9	1.28	0.90	0.65	0.63	0.60	0.35	0.39	0.44
ΣMUFA	25.36	24.12	29.60	27.23	38.16	41.60	42.13	38.08
18:3n−3	0.78	1.11	0.78	0.80	0.46	0.48	0.42	0.43
18:2n−6	15.71	18.78	16.20	15.29	7.51	6.52	5.70	5.30
20:5n−3	1.99	3.27	1.99	3.90	0.66	1.00	1.55	3.34
22:6n−3	5.16	6.97	4.99	9.54	1.08	1.87	2.68	6.24
20:4n6	1.14	2.15	1.36	1.69	0.35	0.41	0.34	0.50
ΣPUFA	24.78	32.29	25.32	31.22	10.06	10.29	10.67	15.81
ΣPUFA n−3	7.93	11.36	7.76	14.24	2.20	3.36	4.64	10.01
ΣPUFA n−6	16.85	20.93	17.55	16.98	7.86	6.93	6.03	5.80
n−3/n−6	0.47	0.54	0.44	0.84	0.28	0.48	0.77	1.73
Total	100.00	100.00	100.00	100.00	100.00	100.00	100.00	100.00

ΣSFA, ΣMUFA, ΣPUFA, Σn−3 PUFA and Σn−6 PUFA are the sum of saturated, monounsaturated, polyunsaturated, polyunsaturated n−3, and polyunsaturated n−6, respectively. n−3/n−6 is the ratio between the sum of n−3 PUFA and n−6 PUFA. PL1—low levels of phospholipids; PL2—high level of phospholipids; TG1—low levels of triglycerides; TG2—high levels of triglycerides.

**Table 3 animals-12-00951-t003:** Nucleotide sequence of primers used for real-time quantitative (qPCR) amplification.

Gene	Primer Sequence 5′ → 3′	R2 (%)	% Efficiency	Genbank Acession Number
Elongase 5	TCACACTCATCTTCCTCTTCTC	0.999	91.66%	KY623454
*(elovl5)*	GGTTTCTCAAATGTCAATCCAC			
Acetyl CoA carboxylase	CATCTTGACTGAACTCACCC	0.999	90180%	KY623451
*(acc)*	CATCCTGACAACCTGATTACTG			
Fatty acid synthase	CTCGCAACTTATTGATGGTG	0.998	96.47%	KY623455
*(fas)*	ATGTAAATAGCCTGAACCCT			
Stearoyl CoA desaturase (delta9)	TCACCAACTATTAGCCACAG	0.999	89145%	KY623452
*(scd)*	TATTGCCTGTAGAGAAACCT			
Carnitine palmitoyltransferase	CAACAATGATCTGCCTTCGT	0.999	94.36%	KY623450
*(cpt)*	CACAAATCACAAACATTCAGCC			
Acyl CoA dehydrogenase (very long chain)	TTGCCATTCTTCAGTTACCA	0.999	92.80%	KY623449
*(acadvl)*	TTTCACTCTTCAACATCTCCA			
Acyl-CoA Oxidase	TTGTTTGTAGACCTCCACCA	0.996	93.59%	KY623453
*(acox)*	ATTGTGTCCTATCTGAATGAAC			
Elongation factor 1 alpha	TGGTACCTCTCAGGCTGAC	0.999	96.87%	-
*(ef1a)*	ACCAAGGGTGAAGGCCAG			

R2—coefficient of correlation.

**Table 4 animals-12-00951-t004:** Effects of dietary DHA and EPA as PL and TG on growth performance, survival, and body parameters of dusky grouper juveniles fed experimental diets for 8 weeks.

Index	Experimental Groups				
	Initial	PL1	PL2	TG1	TG2
Initial body weight (g)		1.43 ± 0.22	1.43 ± 0.22	1.43 ± 0.22	1.43 ± 0.22
Final body weight (g)		8.08 ± 1.74	7.10 ± 1.55	7.75 ± 1.36	7.41 ± 1.78
Initial length (cm)		4.49 ± 0.16	4.49 ± 0.16	4.49 ± 0.16	4.49 ± 0.16
Final length (cm)		7.82 ± 0.59	7.51 ± 0.51	7.80 ± 0.48	7.57 ± 0.54
Weight gain		6.65 ± 1.78	5.67 ± 1.59	6.32 ± 1.34	5.98 ± 1.82
Daily weight gain		0.11 ± 0.02	0.09 ± 0.02	0.10 ± 0.02	0.09 ± 0.03
Liver total lipids (mg/g)	128.90 ± 12.00 ^b^	225.88 ± 66.15 ^a,b^	250.95 ± 61.60 ^a^	121.88 ± 24.48 ^b^	187.45 ± 5.99 ^a^
Muscle total lipids (mg/g)	17.1 ± 2.60 ^b^	16.11 ± 3.49 ^b,c^	21.03 ± 3.53 ^a^	19.13 ± 2.29 ^a^	14.75 ± 1.40 ^c^
Survival (%)		100	100	100	100

PL1—low levels of phospholipids; PL2—high level of phospholipids; TG1—low levels of triglycerides; TG2—high levels of triglycerides. Statistically significant differences between means are indicated with different letters above the means.

**Table 5 animals-12-00951-t005:** Fatty acid percentages of the neutral fractions of the livers of dusky grouper juveniles fed the experimental diets.

FA	Initial	PL1	PL2	TG1	TG2
14:0	7.01 ± 0.21 ^c,b^	6.70 ± 0.35 ^c^	9.48 ± 0.61 ^a^	7.12 ± 0.35 ^c^	7.67 ± 0.65 ^b^
16:0	19.7 ± 0.06 ^b^	22.24 ± 1.44 ^a^	18.65 ± 0.92 ^b^	15.69 ± 0.53 ^c^	15.44 ± 0.75 ^c^
18:0	5.03 ± 0.09 ^a^	4.19 ± 0.25 ^c^	3.65 ± 0.18 ^b^	4.46 ± 0.31 ^c^	4.34 ± 0.68 ^c^
ΣSFA	36.70 ± 0.40 ^a^	35.74 ± 1.89 ^a,b^	36.06 ± 0.11 ^b^	30.63 ± 0.34 ^c^	32.34 ± 0.82 ^d^
16:1n−7	5.03 ± 0.24 ^b^	6.09 ± 0.30 ^a^	4.34 ± 0.16 ^c^	3.12 ± 0.28 ^d^	2.95 ± 0.13 ^d^
18:1n−9	30.13 ± 0.40 ^d^	40.26 ± 1.44 ^b^	46.25 ± 0.48 ^a^	38.34 ± 1.66 ^c^	44.86 ± 1.12 ^a^
ΣMUFA	40.81 ± 0.30 ^d^	51.45 ± 1.91 ^b^	55.32 ± 0.60 ^a^	45.47 ± 2.08 ^c^	52.01 ± 1.02 ^b^
18:2n−6	9.94 ± 0.08 ^a^	7.83 ± 0.44 ^b^	6.24 ± 0.33 ^d^	6.47 ± 0.47 ^d^	7.19 ± 0.31 ^c^
20:5n−3	3.40 ± 0.03 ^b^	1.24 ± 0.13 ^d^	0.52 ± 0.04 ^e^	4.27 ± 0.47 ^a^	2.02 ± 0.08 ^c^
22:6n−3	6.92 ± 0.04 ^b^	2.73 ± 0.37 ^d^	1.14 ± 0.15 ^e^	11.94 ± 1.54 ^a^	5.21 ± 0.39 ^c^
ΣPUFA	22.49 ± 0.09 ^a^	12.81 ± 0.84 ^c^	8.61 ± 0.55 ^d^	23.90 ± 2.19 ^a^	15.65 ± 0.22 ^b^
ΣPUFA n−3	11.94 ± 0.01 ^b^	4.50 ± 0.55 ^d^	2.07 ± 0.22 ^e^	16.67 ± 2.06 ^a^	8.02 ± 0.86 ^c^
ΣPUFA n−6	10.55 ± 0.11 ^a^	8.32 ± 0.46 ^b^	6.54 ± 0.33 ^d^	7.23 ± 0.49 ^c^	7.63 ± 0.66 ^b,c^
Σn−3/Σn−6	1.13 ± 0.01 ^b^	0.54 ± 0.06 ^c^	0.32 ± 0.01 ^d^	2.31 ± 0.28 ^a^	1.06 ± 0.20 ^b^

FA—fatty acids, ΣSFA, ΣMUFA, ΣPUFA, Σn−3 PUFA and Σn−6 PUFA are the sum of saturated, monounsaturated, polyunsaturated, polyunsaturated n−3, and polyunsaturated n−6, respectively. n−3/n−6 is the ratio between the sum of n−3 PUFA and n−6 PUFA. PL1—low levels of phospholipids; PL2—high level of phospholipids; TG1—low levels of triglycerides; TG2—high levels of triglycerides. Statistically significant differences between means are indicated with different letters above the means.

**Table 6 animals-12-00951-t006:** Fatty acid percentages of the polar fractions of the livers of dusky grouper juveniles fed the experimental diets.

FA	Initial	PL1	PL2	TG1	TG2
16:0	16.27 ± 0.73 ^b,c^	15.36 ± 1.18 ^c^	13.45 ± 1.25 ^b^	14.88 ± 1.66 ^b^	18.79 ± 0.69 ^a^
C18:0	20.88 ± 1.13 ^a,b^	19.45 ± 1.86 ^b^	18.44 ± 2.26 ^b^	21.64 ± 1.76 ^a^	6.19 ± 1.42 ^c^
ΣSFA	39.01 ± 1.87 ^b,c,d^	37.14 ± 1.61 ^c^	34.30 ± 0.54 ^d^	40.10 ± 1.69 ^a^	41.85 ± 3.61 ^a^
18:1n−9	11.16 ± 0.71 ^d^	16.34 ± 3.22 ^b^	22.39 ± 2.19 ^b^	17.91 ± 3.47 ^b,c^	34.70 ± 1.79 ^a^
ΣMUFA	14.76 ± 0.82 ^d^	21.89 ± 3.92 ^c^	27.83 ± 2.55 ^b^	22.61 ± 4.22 ^c^	41.03 ± 2.08 ^a^
18:2n−6	11.46 ± 0.36 ^b^	11.25 ± 0.51 ^b^	13.76 ± 0.37 ^a^	5.51 ± 0.34 ^d^	7.49 ± 0.49 ^c^
20:5n−3	5.22 ± 0.21 ^b^	5.02 ± 0.19 ^c^	3.85 ± 0.45 ^d^	6.94 ± 0.82 ^a^	1.94 ± 0.27 ^e^
22:6n−3	24.13 ± 2.88 ^a,b^	19.12 ± 2.21 ^c,e^	14.90 ± 1.35 ^d^	20.39 ± 4.18 ^b,e^	6.17 ± 0.62 ^f^
20:4n−6	4.86 ± 0.60 ^b^	5.08 ± 0.71 ^a^	4.80 ± 0.70 ^b^	4.18 ± 0.54 ^c^	1.09 ± 0.52 ^d^
ΣPUFA	46.23 ± 2.70 ^a^	40.97 ± 2.93 ^b^	37.87 ± 2.09 ^b^	37.29 ± 5.21 ^b^	17.12 ± 1.70 ^c^
ΣPUFA n−3	29.90 ± 3.13 ^a^	24.64 ± 2.29 ^cd^	19.31 ± 1.30 ^c^	27.59 ± 4.95 ^a,b^	8.54 ± 0.71 ^e^
ΣPUFA n−6	16.32 ± 0.42 ^b^	16.33 ± 0.71 ^b^	18.56 ± 1.00 ^a^	9.69 ± 0.37 ^c^	8.58 ± 1.01 ^d^
Σn−3/Σn−6	1.84 ± 0.24 ^b^	1.51 ± 0.09 ^c^	1.04 ± 0.05 ^d^	2.84 ± 0.44 ^a^	1.00 ± 0.04 ^d^

FA—fatty acids, ΣSFA, ΣMUFA, ΣPUFA, Σn−3 PUFA an Σn−6 PUFA are the sum of saturated, monounsaturated, polyunsaturated, polyunsaturated n−3, and polyunsaturated n−6, respectively. n−3/n−6 is the ratio between the sum of n−3 PUFA and n−6 PUFA. PL1—low levels of phospholipids; PL2—high level of phospholipids; TG1—low levels of triglycerides; TG2—high levels of triglycerides. Statistically significant differences between means are indicated with different letters above the means.

**Table 7 animals-12-00951-t007:** Fatty acid percentages of the neutral fractions of the muscles of dusky grouper juveniles fed the experimental diets.

FA	Initial	PL1	PL2	TG1	TG2
12:00	3.29 ± 1.12 ^b^	9.74 ± 1.79 ^a^	11.73 ± 4.31 ^a^	9.68 ± 2.61 ^a^	9.36 ± 2.56 ^a^
14:00	4.86 ± 0.30 ^b^	7.68 ± 0.31 ^a^	8.29 ± 2.61 ^a^	8.23 ± 0.37 ^a^	6.78 ± 1.10 ^a^
16:00	22.68 ± 0.87 ^a^	22.80 ± 0.67 ^a^	17.10 ± 1.37 ^b^	16.02 ± 0.63 ^b^	15.88 ± 0.77 ^b^
18:00	7.15 ± 0.44 ^a^	6.78 ± 0.90 ^a^	7.73 ± 6.78 ^a^	6.31 ± 0.69 ^a^	7.52 ± 1.07 ^a^
ΣSFA	38.0 ± 0.94 ^c^	47.00 ± 1.11 ^a^	44.85 ± 1.66 ^b^	40.24 ± 2.50 ^c^	39.54 ± 2.28 ^c^
18:1n−9	26.13 ± 1.81 ^d^	31.55 ± 0.79 ^c^	38.25 ± 2.03 ^a^	34.82 ± 1.75 ^b^	38.02 ± 1.23 ^a^
ΣMUFA	36.91 ± 1.08 ^d^	38.98 ± 1.20 ^c^	44.24 ± 2.14 ^a^	40.65 ± 1.85 ^b^	43.46 ± 1.08 ^a^
18:2n−6	10.64 ± 0.58 ^a^	8.01 ± 0.41 ^b^	7.07 ± 1.77 ^c^	6.01 ± 0.25 ^d^	6.29 ± 0.19 ^d^
20:5n−3	4.08 ± 0.30 ^a^	1.33 ± 0.13 ^d^	0.71 ± 0.19 ^e^	3.35 ± 0.30 ^b^	2.25 ± 0.46 ^c^
22:6n−3	8.64 ± 0.52 ^a^	3.52 ± 0.62 ^c^	2.18 ± 0.81 ^c^	8.66 ± 0.89 ^b^	7.06 ± 1.58 ^b^
ΣPUFA	25.10 ± 1.30 ^a^	14.02 ± 1.18 ^d^	10.91 ± 3.13 ^e^	19.11 ± 1.29 ^b^	17.00 ± 2.21 ^c^
ΣPUFA n−3	13.67 ± 0.92 ^a^	5.46 ± 0.76 ^c^	3.33 ± 0.94 ^d^	12.47 ± 1.15 ^a^	9.65 ± 1.91 ^b^
ΣPUFA n−6	11.43 ± 0.59 ^a^	8.56 ± 0.48 ^c^	7.58 ± 2.24 ^b,d,e^	6.65 ± 0.21 ^e^	7.34 ± 0.41 ^d^
Σn−3/Σn−6	1.20 ± 0.00 ^b^	0.64 ± 0.06 ^c^	0.44 ± 0.04 ^d^	1.87 ± 0.14 ^a^	1.31 ± 0.22 ^b^

FA—fatty acids, ΣSFA, ΣMUFA, ΣPUFA, Σn−3 PUFA an Σn−6 PUFA are the sum of saturated, monounsaturated, polyunsaturated, polyunsaturated n−3, and polyunsaturated n−6, respectively. n−3/n−6 is the ratio between the sum of n−3 PUFA and n−6 PUFA. PL1—low levels of phospholipids; PL2—high level of phospholipids; TG1—low levels of triglycerides; TG2—high levels of triglycerides. Statistically significant differences between means are indicated with different letters above the means.

**Table 8 animals-12-00951-t008:** Fatty acid percentages of the polar fractions of the muscles of dusky grouper juveniles fed the experimental diets.

FA	Initial	PL1	PL2	TG1	TG2
16:00	20.03 ± 5.35 ^b,c^	28.31 ± 0.62 ^a^	11.83 ± 4.73 ^c^	24.83 ± 2.94 ^b^	23.27 ± 3.84 ^b^
18:00	17.87 ± 3.72 ^b^	34.75 ± 2.03 ^a^	16.35 ± 6.09 ^b^	30.14 ± 4.07 ^a^	16.11 ± 2.72 ^b^
ΣSFA	39.27 ± 8.89 ^c^	64.01 ± 1.99 ^a^	32.80 ± 7.73 ^d^	56.39 ± 1.56 ^b^	41.60 ± 6.56 ^c^
18:1n−9	20.88 ± 4.68 ^b^	25.83 ± 1.40 ^b^	30.04 ± 6.03 ^a,b^	32.72 ± 1.12 ^a^	36.87 ± 1.65 ^a^
ΣMUFA	27.65 ± 5.89 ^b^	31.43 ± 1.66 ^b^	35.98 ± 6.27 ^a^	39.14 ± 1.13 ^a^	42.66 ± 2.31 ^a^
18:2n−6	6.59 ± 0.66 ^a^	1.49 ± 1.06 ^b^	10.61 ± 2.25 ^a^	2.24 ± 0.89 ^b^	7.12 ± 1.92 ^a^
20:5n−3	4.44 ± 2.05 ^a^	0.51 ± 0.25 ^b^	2.99 ± 1.41 ^a^	0.28 ± 0.00 ^b^	1.66 ± 1.51 ^a,b^
22:6n−3	19.14 ± 11.19 ^a^	2.14 ± 0.28 ^b^	14.15 ± 7.95 ^a^	1.63 ± 0.14 ^b^	5.65 ± 4.34 ^ab^
20:4n−6	2.30 ± 1.02 ^a^	0.38 ± 0.21 ^b^	3.04 ± 1.60 ^a^	0.33 ± 0.12 ^b^	1.15 ± 0.80 ^ab^
ΣPUFA	33.08 ± 14.89 ^a^	4.55 ± 0.69 ^c^	31.23 ± 12.80 ^a^	4.48 ± 0.88 ^c^	15.74 ± 8.32 ^b^
ΣPUFA n−3	24.19 ± 13.89 ^a^	2.69 ± 0.48 ^d^	17.57 ± 9.37 ^b^	1.91 ± 0.14 ^e^	7.47 ± 5.95 ^c^
ΣPUFA n−6	8.89 ± 1.89 ^b^	1.86 ± 1.10 ^c^	13.66 ± 3.75 ^a^	2.57 ± 1.00 ^c^	8.27 ± 2.65 ^b^
Σn−3/Σn−6	2.59 ± 1.89 ^a^	1.93 ± 1.06 ^a^	1.19 ± 0.49 ^a^	0.89 ± 0.49 ^b^	0.84 ± 0.45 ^b^

FA—fatty acids; ΣSFA, ΣMUFA, ΣPUFA, Σn−3 PUFA an Σn−6 PUFA are the sum of saturated, monounsaturated, polyunsaturated, polyunsaturated n−3, and polyunsaturated n−6, respectively. n−3/n−6 is the ratio between the sum of n−3 PUFA and n−6 PUFA. PL1—low levels of phospholipids; PL2—high level of phospholipids; TG1—low levels of triglycerides; TG2—high levels of triglycerides. Statistically significant differences between means are indicated with different letters above the means.

## Data Availability

The data presented in this study are available in the article.

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
