# Peer review of "Long-Chain Polyunsaturated Fatty Acids n−3 (n−3 LC-PUFA) as Phospholipids or Triglycerides Influence on Epinephelus marginatus Juvenile Fatty Acid Profile and Liver Morphophysiology"

_animals, 2022, doi:10.3390/ani12080951_

Round 1
Reviewer 1 Report
There are several mistakes/improvements to be correct/done in the manuscript.
1.- Table 1 please use superindex in the list of ingredientes (Deffated Fish Meala instead of Defatted Fish Meala) It is difficult to follow the table in the actual form
2.- Phosphotech is the firm that provided you with the enriched oils (not phosphotec, this is an injectable product with pyrophosphate)
3.- Table 2 is not included in the pdf, only part of it, so the fatty acid profile of the feeds is not available
4.- In all the tables the superindex lettters to show anova statistical significant values are selected randomly. It is better if you put a for the highest value, b for the next high value, and c, d, e,f for the subsequent values, being the last letter for the lowest value.
I.e muscle total lipids from table 1:
PL2 21.03 a, TG1 19.13 a, initial 17.1 b, PL1 16.11 bc, TG2 14.75 c
Once you write the letters in this way it will be easier for the reader to understand the differences among the values. The same can be applied to fig 1 use a for the highest number/difference and then follow with the other letters, i.e Fig 1 f you put b for the highest value and in the other figures you put a. Please change it and use always the same way for applying the letters.
In the discussion you talk about the differences in the accumulation or depletion of fatty acids depending on the fraction of lipids, perhaps a figure showing the differences in total SFA, MUFA, N-3PUFA, N-6PUFA for each fraction and diet in liver and muscle will show much better the results you discuss. Please try to include a histogram showing that.
And also check the English"Diets containing different DHA and EPA levels in PL and TG for did not resulted---" should be did not result (yo are already using past tense)
Author Response
R1 - There are several mistakes/improvements to be correct/done in the manuscript.
1.- Table 1 please use superindex in the list of ingredientes (Deffated Fish Meala instead of Defatted Fish Meala) It is difficult to follow the
table in the actual form
Reply – Thank you for the correction. Please find it modified in the Table 1.
R1 - 2.- Phosphotech is the firm that provided you with the enriched oils (not phosphotec, this is an injectable product with pyrophosphate)
Reply – Thank you for the correction. Please find it modified in the manuscript in Table 1.
R1 - 3.- Table 2 is not included in the pdf, only part of it, so the fatty acid profile of the feeds is not available
Reply – Thank you so much for the information, we kindly ask apologies for that. Probably there was an error in the file. We respectfully
ask your understanding on that and we are providing now the formatted version with the properly inserted table in the manuscript. Please
find it in the page 4.
R1 - 4.- In all the tables the superindex lettters to show anova statistical significant values are selected randomly. It is better if you put a for
the highest value, b for the next high value, and c, d, e,f for the subsequent values, being the last letter for the lowest value.
I.e muscle total lipids from table 1:
PL2 21.03 a, TG1 19.13 a, initial 17.1 b, PL1 16.11 bc, TG2 14.75 c
Once you write the letters in this way it will be easier for the reader to understand the differences among the values. The same can be applied
to fig 1 use a for the highest number/difference and then follow with the other letters, i.e Fig 1 f you put b for the highest value and in the
other figures you put a. Please change it and use always the same way for applying the letters.
Reply – Thank you, we followed the requested modification and all tables and figures were modified according to the reviewer’s suggestions.
2
R1 - In the discussion you talk about the differences in the accumulation or depletion of fatty acids depending on the fraction of lipids,
perhaps a figure showing the differences in total SFA, MUFA, N-3PUFA, N-6PUFA for each fraction and diet in liver and muscle will show
much better the results you discuss. Please try to include a histogram showing that.
Reply – Thank you so much for this very interesting suggestion for the improvement of our work. We do agree that in many cases, histograms
are a very good approach to show the results, but as our focus was to provide n3 LC-PUFA (DHA and EPA) as phospholipids and triglycerides
we separate the diet in phospholipids and triglycerides as well in order to provide to the reader the correct information regarding the amount
of each fatty acid class in each diet. Thus, it will be difficult to provide a histogram in order to evaluate what was incorporated, since both
fractions were ingested by the fish, and they are not shown as total fatty acids in the diets. In our view, both fractions (phospholipids and
triglycerides) can be metabolized and transformed in each other being difficult to assume what was the origin of the incorporated fatty acid.
So, we kindly and respectfully want to ask the reviewers to keep the data as it is presented in the manuscript.
And also check the English"Diets containing different DHA and EPA levels in PL and TG for did not resulted---" should be did not result
(yo are already using past tense)
Reply – Thank you for the correction. It is modified in the text in line….
Reviewer 2 Report
Manuscript ID: animals-1637664
Long-chain polyunsaturated fatty acids n-3 (n-3 LC-PUFA) as phospholipids or triglycerides influence on Epinephelus marginatus juvenile fatty acid profile and liver morphophysiology
- Recommendation
Overview and general recommendation:
Fish are the main sources of nutritionally essential LC-PUFA for human consumption and clarifying the transfer of LC-PUFA from the diet to the fish muscle is critical for the aquaculture sector and fundamental for the efficient and sustainable use of LC-PUFA, a precious and limited marine resource. Furthermore, Fatty Acids (FA) provide energy and affect metabolism/physiology by different mechanisms of action: mainly via lipid mediators and gene transcription factors. The study was carried out in a marine species of commercial interest and provides in experimental diets EPA and DHA in enriched and technologically modified compounds, phospholipids and triglycerides.
Therefore, I recommend that a major overhaul is needed. I explain my concerns in more detail below. I ask authors to specifically address each of my comments in their answer and rewrite the manuscript
I am available to review it again
- Major comments:
2.1. Great variation between FA in experimental diets:
In studies on the incorporation of a specific FA in the carcass, in your case, Long-chain polyunsaturated fatty acids n-3 (n-3 LC-PUFA), the experimental diets must present the smallest source of FA variation between them. Several factors influence the transfer of FA from the diet to the carcass in fish, such as selective incorporation, absorption, β oxidation and the interaction between dietary FA, ​​among others. There is some variation between freshwater and saltwater fish and even variation between species, but it suggests that the actual extent of FA β oxidation in vivo in fish under certain dietary conditions is complex and largely determined by the relative abundance of each FA, but suggests a selective catabolism of C18 FA, following the order: 18: 3n-3 > 18: 2n-6 > 18: 1n-9 and arachidonic acid (ARA, 20: 4n-6), and sparing omega-3 effects of MUFA and SFA mobilization of FA for β oxidation, probably MUFA and SFA > LC-PUFA. For all the physiological importance of EPA and DHA, both are susceptible to β-oxidation when provided in excess in the diet.
2.2. Liver morphophysiology
Two major considerations should be made when evaluating liver lipid deposition in this manuscript. The first is regarding the proportions of Fatty Acids (FA) in the experimental diets, since the inducing effect or not of Hepatic Steatosis (HE) is quite variable between the saturated, monounsaturated and polyunsaturated classes and even alone. Briefly, saturated FAs (SFAs) are described in the literature as the most important and most potent inducers of the lipotoxic effect caused by tissue; however, in the liver, it seems to be less steatogenic than monounsaturated fatty acids, oleic acid, possibly due to its ability to increase the expression of PPARα. It is suggested that polyunsaturated fatty acids (PUFAs), such as α-linolenic, eicosapentaenoic and docosahexaenoic acid, are capable of reversing liver damage and protecting against HE. In the experimental diets, there is no logic of proportion of different classes to be observed, mainly due to the inclusion of two sources of oils and at different levels (coconut oil and olive oil), making it difficult to establish a conclusion about liver morphophysiology.
Another point concerns the methodology adopted in histology. Although the vast majority of studies to assess lipid infiltration in the liver of fish are performed by routine histological methodology (hematoxylin and eosin (H & E), this methodology is not adequate for these studies. In H&E staining, we evaluated the vacuolization, that is, indirect quantification of vacuoles produced by the dissolution of triglyceride droplets in hepatocytes by the solvents used in routine staining. hydropic) or other substances soluble in these solvents. To be sure of the triglyceride content of the vacuoles, special stains such as Sudam red (Sudam III) or black (Sudan IV) and oil red O.
So don't be too emphatic in your conclusions
- Minor comments:
3.1. Deleting phrases referring to feed cost throughout the manuscript does not make sense in the scope of this manuscript.
3.2. Insert brief comments on the importance of PL in the formation of cell membranes and in the formation of plasma lipoproteins
3.3. The composition of the pre-initial diet is missing.
- Suggestions for future work:
4.1. For gene expression analysis, I would exclude some analyzes that in my understanding are not so important for formulated experimental diets, such as elongase 5 (elovl5) and stearoyl CoA desaturase and, I would insert the PPARs transcription factors
4.2. As you are studying the effect of long-chain polyunsaturated fatty acids n-3 (n-3 LC-PUFA) incorporated in two sources, phospholipids or triglycerides, you would perform immunological analysis, plasma lipoproteins and enzymatic bile salt-dependent lipases (triglycerides ) and phospholipase A2, Phospholipase C, di- and mono-acylglycerol lipase (Phospholipids)
4.3. I would work with adult fish, close to slaughter, and not juveniles to evaluate the deposition of n-3 long-chain polyunsaturated fatty acids (n-3 LC-PUFA). We know that the selective incorporation of AG varies with different growth stages.
Author Response
Overview and general recommendation:
Fish are the main sources of nutritionally essential LC-PUFA for human consumption and clarifying
the transfer of LC-PUFA from the diet to the fish muscle is critical for the aquaculture sector and
fundamental for the efficient and sustainable use of LC-PUFA, a precious and limited marine
resource. Furthermore, Fatty Acids (FA) provide energy and affect metabolism/physiology by
different mechanisms of action: mainly via lipid mediators and gene transcription factors. The study
was carried out in a marine species of commercial interest and provides in experimental diets EPA
and DHA in enriched and technologically modified compounds, phospholipids and triglycerides.
Therefore, I recommend that a major overhaul is needed. I explain my concerns in more detail below.
I ask authors to specifically address each of my comments in their answer and rewrite the manuscript
I am available to review it again
R2 - Major comments:
2.1. Great variation between FA in experimental diets:
In studies on the incorporation of a specific FA in the carcass, in your case, Long-chain
polyunsaturated fatty acids n-3 (n-3 LC-PUFA), the experimental diets must present the smallest
source of FA variation between them. Several factors influence the transfer of FA from the diet to the
carcass in fish, such as selective incorporation, absorption, β oxidation and the interaction between
dietary FA, among others. There is some variation between freshwater and saltwater fish and even
variation between species, but it suggests that the actual extent of FA β oxidation in vivo in fish under
certain dietary conditions is complex and largely determined by the relative abundance of each FA,
but suggests a selective catabolism of C18 FA, following the order: 18: 3n-3 > 18: 2n-6 > 18: 1n-9 and
arachidonic acid (ARA, 20: 4n-6), and sparing omega-3 effects of MUFA and SFA mobilization of FA
for β oxidation, probably MUFA and SFA > LC-PUFA. For all the physiological importance of EPA
and DHA, both are susceptible to β-oxidation when provided in excess in the diet.
Reply – Thank you very much for the important consideration and explanation about fatty acids
oxidation preferences in fish metabolism. Indeed, this is correct and also other authors mention that
in their research, for example, Tocher et al. (2003) and Turchini et al. (2009) which mention exactly
this preference order SFA > MUFA > PUFA > LC-PUFA. However, this is highly modulated by several
factors such as species nutritional requirements, diet composition, abiotic factors (mainly
temperature), etc. So, we really appreciate these comments that clarify the interpretations of data.
R2 - 2.2. Liver morphophysiology
Two major considerations should be made when evaluating liver lipid deposition in this manuscript.
The first is regarding the proportions of Fatty Acids (FA) in the experimental diets, since the inducing
effect or not of Hepatic Steatosis (HE) is quite variable between the saturated, monounsaturated and
2
polyunsaturated classes and even alone. Briefly, saturated FAs (SFAs) are described in the literature
as the most important and most potent inducers of the lipotoxic effect caused by tissue; however, in
the liver, it seems to be less steatogenic than monounsaturated fatty acids, oleic acid, possibly due to
its ability to increase the expression of PPARα. It is suggested that polyunsaturated fatty acids
(PUFAs), such as α-linolenic, eicosapentaenoic and docosahexaenoic acid, are capable of reversing
liver damage and protecting against HE. In the experimental diets, there is no logic of proportion of
different classes to be observed, mainly due to the inclusion of two sources of oils and at different
levels (coconut oil and olive oil), making it difficult to establish a conclusion about liver
morphophysiology.
Reply – Indeed all comments about the effects of the fatty acids on the morphophysiology of the liver
are pertinent and highly appreciated, for sure it will improve a lot our future works. It is quite difficult
to evaluate which fatty acid group or even phospholipids and triglycerides had any effect in the
present study since they varied among diets. Anyway, here we tried to use this approach as an
auxiliary tool to evaluate the morphological response of the fish and this data was very useful for the
proposed investigation.
R2 - Another point concerns the methodology adopted in histology. Although the vast majority of
studies to assess lipid infiltration in the liver of fish are performed by routine histological
methodology (hematoxylin and eosin (H & E), this methodology is not adequate for these studies. In
H&E staining, we evaluated the vacuolization, that is, indirect quantification of vacuoles produced
by the dissolution of triglyceride droplets in hepatocytes by the solvents used in routine staining.
hydropic) or other substances soluble in these solvents. To be sure of the triglyceride content of the
vacuoles, special stains such as Sudam red (Sudam III) or black (Sudan IV) and oil red O.
So don't be too emphatic in your conclusions
Reply – Thank you for the valuable suggestions. For sure we will consider these suggested
morphological analysis for futures experiments in the case of conducting liver morphological analysis.
Anyway in this specific study we tried to look at the general pattern of the liver in response to the
diets and not go so deep in this morphological approach. Also, as mentioned by the reviewers some
studies published in reference journals in the aquaculture field used this approach in order to see if
there is any morphological effect in the liver morphology (Kindly see the 2 manuscripts, below;
Araújo et al., 2021; Chen et al., 2020; Marques et al., 2021; Torrencillas et al., 2018). Anyway, we
reconsidered the conclusive comments about the effects of the diet on the morphology of the liver
and we inserted less emphatic comments regarding the liver analysis. But, as mentioned the
comments were highly appreciated and for sure we will consider this valuable suggestion for the
future experiments.
Manuscripts
Araújo, B.C., Rodrigues, M., Honji, R.M., Rombenso, A.N., del Rio-Zaragoza, O.B., Cano, A., Tijanero,
A., Mata-Sotres, J.A., Viana, M.T. 2021. Arachidonic acid modulated lipid metabolism and improved
productive performance of striped bass (Morone saxatilis) juvenile under sub-optimal temperatures.
Aquaculture, 530, 735939.
Chen, Y., Sun, Z., Liang, Z., Xie, Y., Su, J., Luo, Q., Zhu, J., Liu, Q., Han, T., Wang, A. 2020. Effects of
dietary fish oil replacement by soybean oil and L-carnitine supplementation on growth performance,
3
fatty acid composition, lipid metabolism and liver health of juvenile largemouth bass, Micropterus
salmonides. Aquaculture, 516, 734596.
Marques, V.H., Moreira, R.G., Branco, G.S., Honji, R.M., Rombenso, A.N., Viana, M.T., Mello, P.H.,
Mata-Sotres, J.A., Araújo, B.C. 2021. Different saturated and monounsaturated fatty acids levels in
fish oil-free diets to cobia (Rachycentron canadum) juveniles: Effects in growth performance and lipid
metabolism. Aquaculture, 541, 736843.
Torrencillas, S., Betancor, M.B., Caballero, M.J., Rivero, F., Robaina, L., Izquierdo, M., Montero, D.
Supplementation of archidonic acid rich oil in European sea bass juveniles (Dicentrachus labrax) diets:
effects on growth performance, tissue fatty acid profile and lipid metabolism. Fish Physiology and
Biochemistry, 44, 283-300.
R2 - Minor comments:
3.1. Deleting phrases referring to feed cost throughout the manuscript does not make sense in the
scope of this manuscript.
Reply – Accepted and removed from the manuscript.
R2 - 3.2. Insert brief comments on the importance of PL in the formation of cell membranes and in the
formation of plasma lipoproteins
Reply – Accepted and inserted. Please find it in page 2 in line 52.
R2 - 3.3. The composition of the pre-initial diet is missing.
Reply - Indeed this data is not presented, because all fish were fed the same diet and kept at the same
condition prior to starting the experiment.
R2 - Suggestions for future work:
4.1. For gene expression analysis, I would exclude some analyzes that in my understanding are not
so important for formulated experimental diets, such as elongase 5 (elovl5) and stearoyl CoA
desaturase and, I would insert the PPARs transcription factors
Reply – Thank you very much. It will be considered in future studies.
R2 - 4.2. As you are studying the effect of long-chain polyunsaturated fatty acids n-3 (n-3 LC-PUFA)
incorporated in two sources, phospholipids or triglycerides, you would perform immunological
analysis, plasma lipoproteins and enzymatic bile salt-dependent lipases (triglycerides) and
phospholipase A2, Phospholipase C, di- and mono-acylglycerol lipase (Phospholipids)
Reply – Thank you very much for the valuables inputs, for sure it will improve the quality of our
futures studies.
R2 - 4.3. I would work with adult fish, close to slaughter, and not juveniles to evaluate the deposition
of n-3 long-chain polyunsaturated fatty acids (n-3 LC-PUFA). We know that the selective
incorporation of AG varies with different growth stages.
Reply – Yes, this is one of our next aims and will be a great step forward in our research quality to
insert all these valuable suggestions.
Round 2
Reviewer 2 Report
The author made the suggested changes, mainly by not being so emphatic in his conclusions. In the experimental diets, there were variations in the proportion of fatty acids between them, which did not allow to verify the isolated influence of the incorporation of Long-chain polyunsaturated fatty acids n-3 (n-3 LC-PUFA) to PL phospholipids or triglycerides. As for the histological analysis, the methodology was not ideal and the experimental diets presented variations of FA W9, which directly influences hepatic steatosis.